# Diagnostic and Prognostic Role of CD93 in Cardiovascular Disease: A Systematic Review

**DOI:** 10.3390/biom13060910

**Published:** 2023-05-30

**Authors:** Federica Piani, Giovanni Tossetta, Gabriel Cara-Fuentes, Davide Agnoletti, Daniela Marzioni, Claudio Borghi

**Affiliations:** 1Cardiovascular Internal Medicine, IRCCS Azienda Ospedaliero-Universitaria di Bologna, 40138 Bologna, Italy; davide.agnoletti2@unibo.it (D.A.); claudio.borghi@unibo.it (C.B.); 2Department of Medical and Surgical Sciences, University of Bologna, 40138 Bologna, Italy; 3Department of Experimental and Clinical Medicine, Università Politecnica delle Marche, 60126 Ancona, Italy; g.tossetta@pm.univpm.it (G.T.); d.marzioni@staff.univpm.it (D.M.); 4Division of Pediatric Nephrology, Children’s Hospital Colorado, Aurora, CO 80045, USA; gabriel.carafuentes@cuanschutz.edu

**Keywords:** CD93, C1qR1, C1qRp, complement protein 1 q subcomponent receptor, cardiovascular disease, cardiovascular risk factors, cardiovascular mortality, cardiovascular outcomes

## Abstract

Introduction. Cluster of Differentiation (CD) 93 (also known as complement protein 1 q subcomponent receptor C1qR1 or C1qRp) is a transmembrane glycoprotein that can also be present in a soluble (sCD93) form. Recent studies have investigated the role of this protein in cardiovascular disease (CVD). The present systematic review aims to assess the associations between CD93 and cardiovascular (CV) risk factors and disease at both the proteomic and genomic levels. Methods. We conducted systematic searches in the PubMed, EMBASE, and Web of Science databases to identify all human studies since inception to February 2023 that investigated the role of CD93 in CV risk factors, CVD, and CV-associated outcomes. The data collection and analysis have been independently conducted by two reviewers. The search terms included: cardiovascular, heart failure, acute stroke, myocardial infarction, stroke, peripheral artery disease, cardiovascular death, MACE, hypertension, metabolic syndrome, hyperuricemia, diabetes, cd93, c1qr, C1qR1, complement protein 1 q subcomponent receptor. Results. A total of 182 references were identified, and 15 studies investigating the associations between CD93 protein levels or CD93 genetic polymorphisms and the development or prevalence of CV risk factors (i.e., hypertension, dyslipidemia, and obesity) and CVD (i.e., heart failure, coronary artery disease, and ischemic stroke) were included. Although promising, the quality and dimension of the analyzed studies do not allow for a definitive answer to the question of whether CD93 may hold diagnostic and prognostic value in CVD.

## 1. Introduction

Cardiovascular disease (CVD) is the result of vascular endothelial cell dysfunction, which is frequently caused by multiple risk factors such as diabetes, hypertension, and hyperlipidaemia. These factors are the expression of both lifestyle and genetic characteristics [1]. Progress in molecular genetics has resulted in the identification of novel susceptibility genes that have not been previously implicated in the aetiology of CVD, such as Cluster of Differentiation (CD) 93.

CD93, also known as complement protein 1 q subcomponent receptor C1qR1 or C1qRp, is a type I transmembrane glycoprotein prevalently expressed by vascular endothelial cells that can also be cleaved to a soluble form (sCD93) [2]. Although initially studied as a C1q receptor, CD93 has also been shown to exert C1q-independent functions in both physiological and pathological conditions. Moreover, although CD93 is mainly expressed on endothelial cells, its expression was also demonstrated in neurons, cytotrophoblast cells, monocytes, neutrophils, B cells, natural killer cells, and platelets [3,4,5,6,7]. 

CD93 participates in early embryonic stages, immune regulation, including efferocytosis, cellular adhesion, endothelial proliferation and migration, angiogenesis, and lipid metabolism [8,9,10,11]. Most of these functions are crucial for the initiation and development of atherosclerotic processes. An altered lipid metabolism that leads to hyperlipidaemia is the most important risk factor for atherosclerosis. In fact, the primum movens of this disease is the entrapment of oxidised low-density lipoprotein (LDL) in the arterial tunica intima. These lipid particles are phagocyted by macrophages, and in this phase, dysfunction of the innate and adaptive immune responses, including the complement system, may be detrimental and accelerate the atherosclerotic processes [12,13]. Efferocytosis, the ability of macrophages to clean up apoptotic cells, has been shown to play a crucial role in plaque formation, dimension, and stability [13]. Endothelial dysfunction and impaired angiogenesis may also induce the development and progression of atherothrombosis [14]. Since atherosclerosis represents the hallmark sign of CVD and CD93 has been shown to regulate several processes that, if impaired, may induce the initiation or progression of the atheroma, it appears evident why CD93 has gained increasing interest in cardiovascular research. 

The objective of our study is to provide information on the role of CD93 in CVD in the form of a “state-of-the-art” systematic review. Our aim is to review the available literature about the associations between sCD93 levels and CD93 genetic polymorphisms and the incidence of cardiovascular risk factors and CVD, such as heart failure, coronary artery disease, and ischemic stroke.

## 2. Materials and Methods

We performed a systematic search using the databases MEDLINE, Embase, and Web of Science up to February 2023. Two independent reviewers (F.P. and G.T.) screened the articles and evaluated the quality of the included studies. Disagreements were resolved by consensus. Fifteen studies evaluating the associations between CD93 and cardiovascular outcomes were included. A detailed method description follows, with separate descriptions of the search protocol, eligibility criteria, screening process, data extraction, and quality assessment.

### 2.1. Protocol

The protocol for this systematic review was undertaken according to the last Preferred Reporting Items for Systematic Reviews and Meta-analyses protocols (PRISMA-P) guidelines [15].

### 2.2. Eligibility Criteria

The literature search was limited to the English language and human subjects. All types of studies except reviews have been included, namely cross-sectional studies, case-control studies, cohort studies, genome-wide association studies, case reports, and case series. In vitro studies using human biospecimens were also included. Publications concerning the assessment of CD93 in diseases other than CVD were excluded.

The outcome data should include at least one of the following outcomes: arterial hypertension, dyslipidaemia, hyperuricemia, metabolic syndrome, diabetes, myocardial infarction, ischemic stroke, peripheral artery disease, hospitalisation due to cardiovascular causes, and cardiovascular death. Publications were included if sCD93 levels or CD93 genetic polymorphisms were assessed. No restrictions were made regarding the comparison groups or the CD93 measurement methods.

### 2.3. Information Sources and Search Terms

A systematic literature search was performed in the following bibliographic databases: Embase (via Elsevier https://www.embase.com accessed on 7 May 2023), MEDLINE ALL (via Ovid), and Web of Science Core Collection. Additionally, Google Scholar was also searched. These databases were searched from inception until 28 February 2023.

Search terms used in the current study included: cardiovascular, cardiovascular disease, hypertension, dyslipidaemia, hypercholesterolemia, hyperuricemia, metabolic syndrome, diabetes, heart failure, coronary artery disease, myocardial infarction, stroke, ischemic stroke, peripheral artery disease, cardiovascular hospitalisation, cardiovascular death, cardiovascular mortality, MACE, CD93, C1qr, C1qR1, complement protein 1 q subcomponent receptor.

### 2.4. Study Selection, Data Extraction, and Quality Assessment

All articles were imported into EndNote X9. After removal of duplicates, two reviewers (F.P. and G.T.) independently screened the titles, abstracts, and full text of the records for eligibility. Disagreements were resolved by consensus. F.P. and G.T. extracted relevant information for each outcome from the included articles. A description of participants and interventions for each study is provided throughout the text and in Table 1. No randomised, controlled studies were found. The methodological quality and risk of bias of the selected studies were evaluated according to the STROBE standard [16]. In particular, reviewed studies were evaluated by the 22-item STROBE checklist. The scoring was performed independently by 2 authors (F.P. and G.T.), and their eventual disagreements were resolved by discussion and consensus. Studies that presented from 10 to 16 criteria out of 22 were evaluated as having medium-quality and medium-risk bias; studies above and below this threshold were considered high-quality and low-quality, respectively. The overall quality of the included studies was moderate, and the number of cases was low to moderate except for genetic studies, which overall included a high number of cases. 

## 3. Results

A total of 182 references were identified (37 from Embase, 137 from Medline, and 8 from Web of Science) and imported into EndNote X9. After removing 21 duplicates, 161 articles were eligible for title and abstract screening. During this phase, 129 articles were excluded since they did not assess sCD93 levels or CD93 polymorphisms. Thirty-two articles were eligible for full-text reading. Then, 17 articles were excluded after full-text reading because no direct association was investigated between CD93 and the cardiovascular outcomes listed in the eligibility criteria (Figure 1).

In total, 15 articles fulfilled the inclusion criteria. No articles were found for the following outcomes: hyperuricemia and peripheral artery disease. 

## 4. Discussion

The role of CD93 in health and CVD is summarised in Figure 2. In the upcoming text, we provide a summary of the current literature on the role of CD93 in the main outcomes under study. 

In each paragraph, we will summarise the evidence from the included studies. More detailed information regarding each study’s results is provided in Table 1.

### 4.1. CD93 and Lipid Metabolism

Although the detailed molecular mechanisms of the association between CD93 and lipid profiles have not been clarified, in the present study, we found three papers that investigated this association. 

In the study by Van der Net and colleagues, 2145 patients with familial hypercholesterolemia (FH) were genotyped to assess the associations between 10 genetic polymorphisms and the risk of developing coronary heart disease [17]. Only four out of 10 studied polymorphisms were associated with an increased risk of cardiovascular outcome, and one of those was CD93 gene polymorphism rs3746731. Although the authors found no associations between the CD93 gene polymorphism and the lipid profile of participants, the fact that this CD93 polymorphism was associated with the development of coronary artery disease in a cohort of patients with FH might suggest that this gene plays a role in the further increased CV risk in this specific dyslipidemic population. 

The expression of CD93 in monocytes and macrophages was associated with atherosclerosis in both animal and human studies [11,31]. Macrophages play a central role in the very first phases of atherosclerosis, regulating the inflammation levels in response to the lipid accumulation in the arterial tunica intima and phagocyting oxidised LDL particles and cholesterol crystals [32]. A study by Patel and colleagues examined the associations between circulating monocyte phenotypes and lipid profiles in 30 individuals with heterogenous lipid profiles [11]. The number of all monocyte phenotypes expressing CD93, namely the precursors of the M2 macrophage subpopulation, was associated with ApoB levels. Furthermore, each different monocyte phenotype (classical, intermediate, and non-classical) was differentially associated with the ApoA1/ApoB ratio, the Total Cholesterol/HDL-C ratio, and HDL-C. Specifically, in non-classical monocytes, there was a negative association with the ApoA1/ApoB ratio; in intermediate monocytes, CD93 expression was positively associated with the Total Cholesterol/HDL-C ratio and negatively correlated with HDL-C. 

Lipid metabolism is known to regulate macrophage differentiation into different phenotypes with diverse functions [33]. One of those functions is efferocytosis, a crucial mechanism in the reduction of inflammation in atherosclerosis [34]. Interestingly, efferocytosis has been shown to be regulated by CD93 [35]. In turn, macrophage phenotypes and functions regulate systemic inflammation with direct effects on lipid metabolism [34]. Indeed, chronic inflammatory states are associated with decreased serum levels of HDL and increased levels of triglycerides [36]. Corroborating this last hypothesis, in the study by Strawbridge and colleagues, serum sCD93 levels were directly associated with HDL cholesterol and negatively correlated with triglyceride levels in the cross-sectional analysis of lipid-lowering-naive healthy subjects [22]. Future controlled studies may clarify whether CD93 could represent a novel target for anti-inflammatory therapeutic strategies in CVD. 

### 4.2. CD93 and Diabetes

There are now several pieces of evidence suggesting an association between CD93 and diabetes. 

In the longitudinal genome-wide study by Chan and colleagues on 15,346 women of different ethnicities, CD93 ranked among the top 10 key driver genes commonly shared between CVD and type 2 diabetes [20]. 

In the study of Strawbridge et al., 901 participants with type 2 diabetes and more than 2000 controls were examined cross-sectionally to assess the differences in sCD93 and the longitudinal associations with type 2 diabetes development and intima-media thickness (IMT) progression as a marker of atherosclerosis [22]. The authors found that levels of sCD93 were significantly lower in subjects with type 2 diabetes compared with healthy controls. Furthermore, subjects who did not progress to prediabetes or diabetes had significantly higher values of sCD93 at baseline. On the contrary, sCD93 did not correlate with IMT progression. Finally, hyperglycaemia induced an impairment in the stress-induced release of CD93. Altogether, these findings seem to suggest that CD93 may play a central role in glucose tolerance, insulin release, and peripheral insulin sensitivity. 

As for diabetes-associated complications, one published study [25] and a pre-print study [37] cross-sectionally evaluated serum sCD93 levels and markers of kidney function (i.e., serum creatinine, eGFR, proteinuria, and urinary albumin to creatinine ratio) in subjects with type 2 diabetes. The pre-print study also included healthy controls. In both studies, higher sCD93 values were associated with a higher urinary albumin-to-creatinine ratio and lower eGFR values. These results might suggest that serum sCD93 may serve as a prognostic marker in diabetic kidney disease. Future studies in prediabetic and diabetic subjects are warranted to clarify the potential role of this biomarker in the initiation and progression of diabetes-associated microvascular complications. 

### 4.3. CD93 and Arterial Hypertension

Hypertension is the leading modifiable cause of CV death and one of the global targets of the World Health Organization (WHO) for preventive strategies. According to the WHO, over 80% of adults with arterial hypertension do not have their blood pressure under control, with one in six patients having resistant hypertension [38]. It is thus evident how novel therapeutic targets for the treatment of this CV risk factor could be extremely helpful. In the present study, we found three articles investigating the associations between CD93 and arterial hypertension. 

A large study by Alehagen and colleagues on a longitudinal cohort of healthy community-living elderly from Sweden showed that the participants in the highest quartile of serum sCD93 had an increased prevalence of arterial hypertension (82% vs. 68% in the remaining quartiles, *p* = 0.011) [24].

In the previously mentioned study by Van der Net et al. on patients with FH, the authors found an association between the rs3746731 CD93 genetic polymorphism and the presence of hypertension (*p* = 0.02) [17].

Finally, a study on the associations between genetic variants of Thrombospondin-1 Levels and high-altitude hypertension found significant associations between CD93 polymorphisms and arterial hypertension in a study sub-group of 349 people living at high altitude (>3500 mt) [26]. The study by Sharma and colleagues found that two intergenic single nucleotide polymorphisms (SNPs) that regulate both thrombomodulin and CD93 levels, and CD93 SNPs rs7492 and rs2749812, were associated with hypertension at high altitude [26]. The condition of hypobaric hypoxia to which highlanders are subjected not only leads to arterial hypertension but also to pulmonary hypertension [39]. Interestingly, a case-control study by Helleberg and colleagues showed that patients with pulmonary hypertension and left heart failure had higher levels of serum sCD93 compared with healthy controls (*p* < 0.001) [29]. Although this difference in sCD93 may be explained by the concomitant presence of heart failure (see below), it can be hypothesised that CD93 plays a role in the regulation of both systemic and pulmonary hypertension. Further validation of these results and these hypotheses may help identify contributing factors to the pathophysiology of hypertension and potential targets for novel therapeutic strategies.

### 4.4. CD93 and Obesity

Obesity has become increasingly prevalent in recent decades, representing a major risk factor for cardiovascular mortality and morbidity [40]. Excessive adipose tissue could lead to non-alcoholic fatty liver disease that, together with adipocyte-released hormones and other signalling molecules, contributes to the development of insulin resistance and, ultimately, to type 2 diabetes [41]. Corroborating this pathophysiologic mechanism, in the study by Strawbridge and colleagues on subjects with type 2 diabetes and controls, sCD93 levels were inversely associated with BMI, insulin, and the HOMA index and positively correlated with adiponectin levels [22]. Another recent study by Snelder and colleagues seems to confirm the association between CD93 and obesity [30]. One year after bariatric surgery, sCD93 levels significantly decreased compared with pre-surgery levels in 72 obese patients. On the contrary, in the small study (n = 97) by Lee and colleagues, there were no significant differences in BMI between high and low sCD93 groups [25]. Further research is needed to better understand the complex interplay between CD93 and obesity and develop novel treatments to fight this highly prevalent and morbid clinical condition.

### 4.5. CD93 and Hyperuricemia

Although we did not identify any studies on this topic, it is well known that hyperuricemia is associated with hypertension, obesity, and insulin resistance [42,43,44]. Thus, it would not be surprising if future studies found an association between CD93 and serum uric acid levels.

### 4.6. CD93 and Coronary Artery Disease

Coronary artery disease remains one of the leading causes of morbidity and mortality worldwide [45]. This disease is the result of the effects of one or more cardiovascular risk factors and eventually leads to myocardial infarction, a major cardiovascular outcome. The human studies we found that first observed an association between CD93 and cardiovascular disease address specifically coronary artery disease and date back to 2008 [17]. As seen in the previous sections, the study by Van der Net analysed 10 different genetic polymorphisms in patients with heterozygous FH [17]. Interestingly, only four out of ten polymorphisms were associated with an increased risk of coronary heart disease. One of those was the rs3746731 polymorphism in the CD93 gene. The rs3746731 polymorphism of the CD93 gene was associated with a 1.26-times increased risk of developing coronary heart disease independently of cardiovascular risk factors such as hypertension, diabetes, and obesity. Furthermore, as the rs3746731 polymorphism is recessively inherited, patients homozygous for the T-allele of the CD93 polymorphism had a 26% increased risk of coronary heart disease compared with patients with at least one C-allele of that polymorphism (*p* = 0.01) [17]. A possible explanation for this association lies in the involvement of CD93 in atherosclerotic processes, as previously described. Another mechanism by which CD93 may play a role in coronary artery disease is its role in endothelial homeostasis and microvasculature function [46,47]. Corroborating this hypothesis, in the study by Chandramouli et al., sCD93 levels were associated with coronary microvascular dysfunction in men with heart failure at preserved ejection fraction [28]. However, the authors did not find a significant association in the female population with the same conditions. Regarding myocardial infarction, we found two studies with contrasting results. In the population study of Mälarstig and colleagues (n = 765), different sCD93 tertiles had similar risks of myocardial infarction in a subgroup of 316 study participants [18]. On the contrary, in the case-control study by Youn et al. of 120 patients with myocardial infarction and 120 matched controls, sCD93 levels were significantly higher in patients with acute myocardial infarction than controls (*p* < 0.0001). Furthermore, as we will discuss in the appropriate section, sCD93 levels were predictive of cardiovascular and all-cause mortality [21]. Both studies used an ELISA assay to measure circulating sCD93; however, in the study by Mälarstig, they used plasma samples, while in Youn’s study, serum samples were used. Serum and plasma sCD93 concentrations should not significantly differ; however, differences in other complement components and receptors have been previously observed in serum vs. plasma [48]. Both studies seem to adjust results for possible confounders, although sample size differs. Future studies may clarify the reasons for these conflicting results. However, it must be highlighted that two out of three studies support the association between CD93 and coronary artery disease.

### 4.7. CD93 and Heart Failure

Heart failure has been the primary cause of hospitalisation in the elderly for many years [49]. We found three papers on the association between CD93 and heart failure. The first study was performed in 2019 by Bouwens and colleagues [23]. The authors investigated the impact of serum sCD93 levels and the risk of a primary composite outcome (cardiovascular mortality, heart failure hospitalisation, heart transplantation, or left ventricular assist device implantation) in 263 patients affected by heart failure. In patients who met the outcome, sCD93 was higher both at baseline and at follow-up (~2.2 years). Additionally, the temporal CD93 trajectory was strongly and independently associated with clinical outcome in patients with chronic heart failure, with an HR of 1.43 per 0.1 standard deviation increase of the annual slope of CD93 at any timepoint during follow-up. The genetic study of Alehagen and colleagues supports the results of the former one. In fact, they found that higher plasma concentrations of the CD93 gene’s polymorphism rs2749812 were associated with higher concentrations of N-terminal pro-B-type natriuretic peptide (NT-proBNP), a marker of the severity of heart failure-associated congestion [24]. Finally, in the case-control study by Helleberg et al., in patients with left heart failure and pulmonary hypertension who underwent heart transplant surgery, sCD93 levels after surgery were similar to those of healthy controls. On the contrary, patients in the same conditions without heart transplantation had higher sCD93 levels compared with healthy controls. Although further studies are needed to definitely assess the role of CD93 in heart failure, current evidence suggests that this protein may also represent a molecular target in this cardiovascular disease. 

### 4.8. CD93 and Stroke

Stroke is the second-leading cardiovascular cause of disability and death worldwide [50]. In the present study, we found two papers on the associations between ischemic stroke and CD93. No studies on haemorrhagic strokes or CD93 were found. In a small cross-sectional case-control study on 33 individuals with ischemic stroke and matched controls, the CD93 transcript was 2-times higher in patients who suffered from ischemic stroke [19]. A longitudinal study on 320 patients presenting an ischemic stroke in the previous 72 h demonstrated that CD93 was among the 16 key biomarkers out of a total of 92 blood biomarkers studied that were able to predict stroke-associated mortality [27]. The authors found that higher sCD93 levels were associated with an almost 4-time higher risk of mortality at 90 days after acute ischemic stroke. Importantly, the association remained significant in multivariable analysis. As previously observed for other clinical conditions, the association between CD93 and stroke needs to be validated in future longitudinal controlled studies that are adequately powered to define the prognostic role of CD93 in stroke.

### 4.9. CD93 and Peripheral Artery Disease

No studies were found for this outcome.

### 4.10. CD93 and CV Mortality

Bicvic et al. demonstrated that circulating serum sCD93 levels were associated with mortality after an acute ischemic stroke [27]. In the study by Youn and colleagues, serum sCD93 baseline levels were an independent predictor of all cause and cardiovascular mortality after adjustment for age and left ventricular ejection fraction at a mean follow-up of 7 months after an episode of acute myocardial infarction [21]. On the contrary, in the population study by Alehagen and colleagues, the authors did find an association between plasma concentrations of CD93 rs2749812 and all-cause mortality but not with cardiovascular mortality [24].

## 5. Conclusions

Cardiovascular disease severely impacts global health, imposing a significant economic burden. Despite optimal treatment, there is still a significant residual number of patients experiencing major events. Treatments addressing novel molecular and epigenetic targets are urgently needed. In the present study, we summarise current evidence on the potential role of CD93 in a broad spectrum of cardiovascular clinical conditions. From the results of our review, CD93 holds promise to become a novel target for cardiovascular risk assessment and prevention. Although encouraging, current evidence does not allow us to conclusively confirm the diagnostic and prognostic value of this novel biomarker in cardiovascular disease. Future studies are warranted to answer the question of whether CD93 may hold prognostic value and if direct or RNA-based therapeutics are warranted.

## Figures and Tables

**Figure 1 biomolecules-13-00910-f001:**
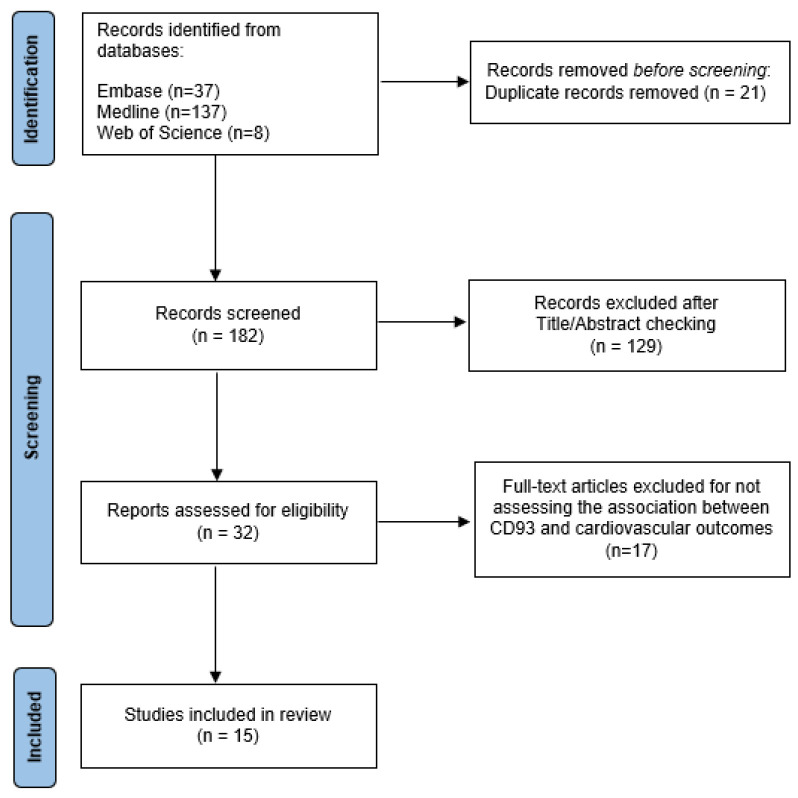
Flow diagram illustrating the article selection process of this review.

**Figure 2 biomolecules-13-00910-f002:**
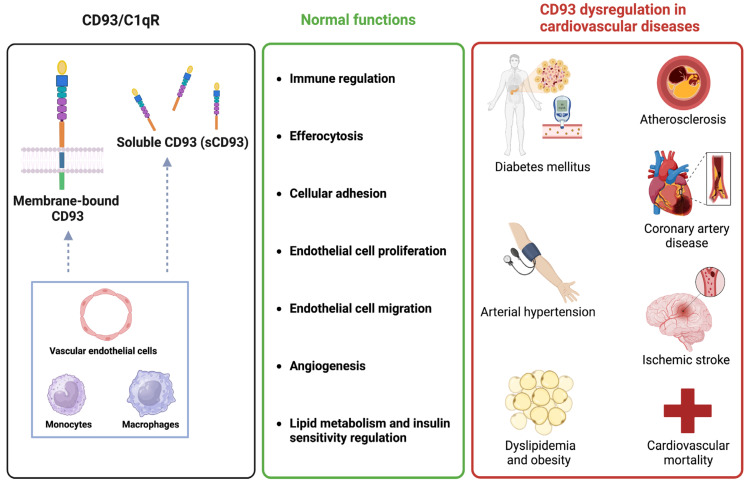
CD93 function and its associations with cardiovascular diseases. In the left black box, CD93 is illustrated with its domains in either membrane-bound or soluble form. CD93 is mainly expressed by vascular endothelial cells, monocytes, and macrophages and mediates multiple functions in physiology, as shown in the middle green box. The associations reported in the present systematic review between CD93 and the risk of diverse cardiovascular diseases are illustrated in the right red box. Created with www.biorender.com.

**Table 1 biomolecules-13-00910-t001:** Summarising table of the fifteen included studies.

	Population	CD93 Measurement	Outcome	Results
Van der Net J.B. et al., 2008 [17]	Cohort study of 2145 patients with heterozygous familial hypercholesterolemia	Genotyping of CD93 rs3746731 polymorphism	Longitudinal study with a mean follow-up of 5 years and an outcome of the development of coronary heart disease.	The rs3746731 polymorphism was associated with an increased risk of coronary heart disease (HR 1.26, 95% CI 1.06–1.49, *p* = 0.01). Additional adjustments for CV risk factors yielded similar results. Patients homozygous for the T-allele of the CD93 polymorphism had a further increase in the risk of coronary artery disease (*p* = 0.01).
Mälarstig A. et al., 2011 [18]	Population study of 764 individuals of both sexes (predominantly men) with median age 54 years	ELISA assay for plasma sCD93	Cross-sectional differences in the whole population plus the risk of premature MI in 316 individuals (159 had premature MI and 157 did not).	In the cross-sectional analysis, elevated levels of sCD93 were associated with increased plasma fibrinogen (*p* = 0.03) and cystatin-C (*p* < 0.001). No significant associations with the risk of MI were observed among different sCD93 tertiles after adjustment. In the unadjusted analysis, the middle tertile was associated with a decreased risk of MI.
Adamski M.G. et al., 2014 [19]	Case-control study (18 ischemic stroke patients vs. 15 matched controls)	High-throughput next-generation qPCR for CD93 gene transcript	Cross-sectional comparison of 40 transcripts between patients with ischemic stroke and controls.	CD93 transcript was 2-times higher in patients who suffered from ischemic stroke compared with control subjects (adjusted *p* value = 0.03).
Chan K.H. et al., 2014 [20]	Longitudinal genome-wide study on 15,346 women of different ethnicities	Genome-wide genotyping (Affymetrix and Illumina), standard SNP analysis	CVD development (MI, stroke, deep vein thrombosis, and pulmonary embolism) and/or type 2 diabetes development.	CD93 ranked among the top 10 key driver genes, which, when perturbed, can potentially affect a large number of genes involved in cardiovascular disease and type 2 diabetes pathways.
Youn J.-C. et al., 2014 [21]	120 patients with acute MI and 120 matched controls	ELISA assay for serum sCD93	Cross-sectional differences plus all-cause and CV deaths in a mean follow-up of 7 months.	sCD93 was significantly higher in patients with acute MI (552.1 ± 293.7 vs. 429.8 ± 114.2 ng/mL, *p* < 0.0001). Initial sCD93 level was an independent predictor of all-cause (*p* = 0.002) and CV mortality (*p* = 0.033) when controlled for age and EF.
Strawbridge R.J. et al., 2016 [22]	Different cohorts of subjects with type 2 diabetes (n = 901) and subjects without diabetes (n = 2470)	Quantification of blood sCD93 by Meso Scale platform and SECTOR Imager 2400	Cross-sectional differences of diabetic vs. non-diabetic subjects and associations with IMT modifications (follow-up at 15 and 30 months) or development of T2D or glucose intolerance (follow-up of 8–10 years).	Levels of sCD93 were significantly lower in subjects with T2D (157 ± 40 vs. 164 ± 45 ng/mL, *p* < 0.0001) and higher in subjects who remained normal glucose tolerant at follow-up (166 ± 44 vs. 158 ± 45 ng/mL, respectively; *p* = 0.016). In lipid-lowering-naive subjects, sCD93 levels were positively associated with HDL, adiponectin, and vitamin D and negatively with TG, BMI, insulin, and HOMA. sCD93 levels at baseline were not associated with any baseline or progression measures of IMT.
Patel V.K. et al., 2017 [11]	Healthy individuals with heterogenous lipid profiles (n = 30)	Whole blood flow cytometry for monocyte expression of CD93	Cross-sectional study to assess the associations between monocyte phenotype and lipid profile.	In all monocyte types, CD93 expression was positively associated with ApoB levels. CD93 in non-classical monocytes is negatively associated with ApoA1/ApoB ratio (R = −0.445, *p* = 0.049); in intermediate monocytes, it is positively associated with TC/HDL-C ratio (R = 0.459, *p* = 0.042) and negatively correlated with HDL-C (R = −0.457, *p* = 0.043).
Bouwens E. et al., 2019 [23]	263 patients with HF	Serum sCD93 by high-throughput proximity extension assays (Olink Proteomics Cardiovascular III multiplex assay)	Composite outcome: CV mortality, HF-hospitalisation, heart transplantation, or LVAD implantation. Median follow-up of 2.2 years.	In patients who met the primary outcome, sCD93 was higher both at baseline and at follow-up. Adjusted HR for the primary outcome per 0.1 standard deviation of the annual slope of CD93 at any point in time during follow-up (estimated by joint modelling analysis) was 1.43 (CI 1.13–1.92, *p* = 0.002).
Alehagen U. et al., 2020 [24]	Population study of 457 individuals of both sexes aged between 70 and 85 years	PCR quantification of CD93 SNP rs2749812 in plasma	Cross-sectional differences plus all-cause and cardiovascular deaths in a mean follow-up of 6.7 years.	CD93 rs2749812 was associated with higher concentration of plasma NT-proBNP (*p* = 0.03). People in the highest quartile of CD93 rs2749812 showed a significant increase in all-cause mortality (*p* = 0.037) but not in CV mortality (*p* = 0.43).
Lee M. et al., 2020 [25]	97 subjects with T2D (mean age 56 years)	ELISA assay for serum sCD93	Cross-sectional analysis of the associations between sCD93 levels, participant characteristics, and markers of diabetic micro- and macrovascular complications.	Higher sCD93 levels were associated with higher proteinuria and ACR values and lower eGFR values. After adjustment for multiple confounders, sCD92 was still associated with a decreased eGFR (β = −14.734, standard error (SE) = 5.564, *p* = 0.010) and an increased ACR (β = 387.943, SE = 191.129, *p* = 0.046).
Sharma K. et al., 2021 [26]	Case–control study of 349 subjects (hypertensive vs. controls) living at high-altitude (≥3500 mt)	Genotype analysis of SNPs by SNPs Fluidigm 48.48 SNPtype assay	Cross-sectional study to investigate the associations between genetic variants of THBS-CD Family Genes and hypertension at high altitude.	Intergenic SNPs rs1998081 and rs2424515 of thrombomodulin and CD93, and CD93 SNPs rs7492 and rs2749812 were associated with the presence of hypertension at high-altitude.
Bicvic A. et al., 2022 [27]	320 patients who had an ischemic stroke within 72 h	Serum sCD93 by high-throughput proximity extension assays (Olink Proteomics multiplex assay)	Longitudinal study on the associations between 92 biomarkers and mortality from any cause within 90 days after stroke.	sCD93 was significantly associated with increased mortality at 90 days after suffering an ischemic stroke (OR 4.30, CI 1.99–9.30; *p* value 0.0002).
Chandramouli C. et al., 2022 [28]	182 patients of both sexes with HFpEF (mean age 74.2 years)	Serum sCD93 by high-throughput proximity extension assays (Olink Proseek Multiplexcardiovascular II and III)	Cross-sectional study on sex-related differences in biomarkers associated with coronary microvascular dysfunction.	In lasso-penalized regression analyses, sCD93 was associated with coronary microvascular dysfunction in men but not in women.
Helleberg S. et al., 2022 [29]	Case-control study (20 healthy controls, 67 patients with left HF and pulmonary hypertension, and 19 who underwent heart transplantation)	Serum sCD93 by high-throughput proximity extension assays (Olink Proteomics, Proseek Multiplex immunoassay reagent kits)	Cross-sectional and longitudinal study on differences in 67 inflammatory proteins in controls vs. left HF and pulmonary hypertension and in a subgroup before and after heart transplantation.	Patients with left HF and pulmonary hypertension showed higher levels of sCD93 compared with healthy controls (*p* < 0.0001). Patients who underwent heart transplantation had lower levels of sCD93 compared with the pre-transplantation timepoint (*p* < 0.0001). sCD93 levels after heart transplant were similar to those of healthy controls.
Snelder S.M. et al., 2022 [30]	72 obese patients who underwent bariatric surgery	Serum sCD93 by high-throughput proximity extension assays (Olink Proteomics Cardiovascular III multiplex assay)	Longitudinal study to investigate cardiovascular biomarkers pre- and after bariatric surgery and their predictive ability for cardiac dysfunction reversal after surgery.	sCD93 levels significantly diminished from before to one year after bariatric surgery. Although not significant, a trend towards decreased levels of sCD93 in patients who had a reversal of pre-surgery cardiac dysfunction was observed (2200 vs. 2572, *p* = 0.22).

Table 1. In the first column are reported the surname and initials of the first authors and the years of publication of the included studies. In the second column, the numerosity and characteristics of the study populations are listed. In the third column, we described the methods of CD93 assessment. In the fourth column, we reported the study design, aims, and outcomes. In the last column, we summarised the results of the included studies as regards the association between CD93 and cardiovascular outcomes. List of abbreviations: ACR, urinary albumin to creatinine ratio; CV, cardiovascular; CVD, cardiovascular disease; DKD, diabetic kidney disease; eGFR, estimated glomerular filtration rate; LVAD, left ventricular assist device; HF, heart failure; HFpEF, heart failure with preserved ejection fraction; HOMA, Homeostatic Model Assessment; HR, hazard ratio; IMT, intima-media thickness; HDL-C, high-density lipoprotein cholesterol; MI, myocardial infarction; T2D, type 2 diabetes; TC, total cholesterol; TG, triglycerides.

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
