# Peer review of "Diagnostic and Prognostic Role of CD93 in Cardiovascular Disease: A Systematic Review"

_biomolecules, 2023, doi:10.3390/biom13060910_

Round 1

Reviewer 1 Report

The authors present a systematic review in relation to de CD93 in cardiovascular disease. The work is correctly done, shows interesting results and the conclusions are correct. CD93 can be relevant in different aspects of human diseases but the level of information and evidence is not clear.

The main concerns are:

-          The authors have excluded 129 papers but the authors do not indicate any criteria and there is no an evaluation of each paper included

-          There is no a synthesis methodology description

-          There is no a description of the criteria used to interpret the level of relevance, reliability or the importance/significance analysis of the papers included.

-          The number of articles indicated in the first paragraph of results does not agree with figure 1: 182 papers, 21 duplicates, 129 excluded and 32 for full-text, but in the text 93 articles were eligible for abstract screening. The criteria in each step should be clarified.

Reviewer 2 Report

Review presents data about the associations between CD93 and cardiovascular risk factors and cardiovascular disease. From this point is the review focused on an actual and important topic.     I have some comments:

1. In lines 121-122 you wrote: “In the upcoming text, we provide a summary of the current literature on the role of CD93 in the main outcomes under study, Figure 2.”

In Figure 2 is not presented summary of current literature. There is presented function of CD93 and its association with cardiovascular diseaases.

2. The legend and more detail information to the Table 1 is missing.

3. The outcomes and in particular results in the Table 1 are presented at great length.

The data could be presented more informative.

4. Lines 153-155: “Furthermore, each different monocyte phenotype (classical, intermediate, and non-classical) was differentially associated with ApoA1/ApoB ratio, Total Cholesterol/HDL-C ratio, and HDL-C (for further details see Table 1).”

The details should be presented here and not in Table 1.

Round 2

Reviewer 1 Report

The authors have answered all the comments.